# The Assessment of Renal Functional Reserve in β-Thalassemia Major Patients by an Innovative Ultrasound and Doppler Technique: A Pilot Study

**DOI:** 10.3390/jcm11226752

**Published:** 2022-11-15

**Authors:** Federico Nalesso, Matteo Rigato, Irene Cirella, Maria Paola Protti, Ruggero Zanella, Bartolomeo Rossi, Maria Caterina Putti, Francesca K. Martino, Lorenzo A. Calò

**Affiliations:** 1Department of Medicine, University of Padova, 35128 Padova, Italy; 2Haematology-Oncology Clinic, Women and Child’s Health Department, University of Padua, 35122 Padua, Italy

**Keywords:** renal functional reserve, beta-thalassemia, ferritin, iron overload, ultrasound, Doppler, renal resistive index, tubulopathy, acute kidney injury, oral protein load

## Abstract

Beta-thalassemia syndromes are the most common inherited monogenic disorders worldwide. The most common pathophysiologic and clinical renal disease manifestations of in β-TM patients is the tubular dysfunctions related to iron overload, chronic anemia, and the need for chronic iron chelation therapy. The aim of this pilot study is to apply an innovative ultrasound and Doppler technique to assess the Renal Functional Reserve (RFR) in β-TM patients, and to evaluate its reliability in iron overload tubulopathy. Ultrasound assessment of intra-parenchymal renal resistive index variation (IRRIV) has recently been proposed as a safe and reproducible technique to identify RFR presence. We define the preserved RFR when the Delta Renal Resistive Index (RRI) is >0.05 (baseline RRI—minimum RRI value during stress) in the Renal Stress Test (RST). Nineteen β-TM patients were enrolled for this study. In our series, we found a strong negative correlation between mean ferritin values and Delta RRI (R = −0.51, *p* = 0.03). This pilot study suggested the RST as reliable tool for assessing the RFR by ultrasound. Specifically, RST could help in clinical practice suggesting the patient’s management and iron chelation therapy.

## 1. Introduction

Hemoglobinopathies are the most common inherited monogenic disorders worldwide. ß-thalassemias are heterogeneous disorders caused by reduced or absent beta-globin synthesis, leading to an imbalance of the globin chains [1].

β-thalassemia is associated with oxidative stress, ineffective erythropoiesis and chronic hemolytic anemia with compensatory hemopoietic expansion [2,3,4]; in this condition, transfusions of packed red blood cells are a mainstay of treatment for thalassemia [5].

In this clinical condition, iron overload is a consequence and, accordingly, iron chelation is needed to prevent or reverse iron overload. Currently, three iron chelators are available: oral agents deferasirox (DFX) and deferiprone (DFP), and the parenteral deferoxamine mesylate (DFO) [6]. This iron chelation therapy has led to a notable improvement in the patients’ prognosis although some patients present dysfunctions in several organs, including renal complications [7], tubular dysfunctions that have been related to iron overload, chronic anemia, and iron chelation therapy [8,9]. In clinical practice, the chelation therapy has to be monitored over time in terms of efficacy and possible nephrotoxic effects such as tubulopathy and acute kidney injury (AKI).

In β-thalassemia, chronic anemia reduces the systemic vascular resistance that leads to hyperdynamic circulation and higher plasma flow in the kidneys with increased glomerular filtration rate (GFR) [10]. Over time, these hemodynamic changes in association with tubular damage can lead to a progressive GFR decline through the typical pathway of hyperfiltration—albuminuria—progressive renal injury, leading to the progression of Chronic Kidney Disease (CKD) [11,12]; this leads to a gradual reduction in renal functional reserve (RFR) despite having normal creatinine level at the lab tests. During phases of glomerular hyperfiltration at least 50% of the nephrons are lost before clinical renal damage is identified by an increase in creatinine level, hence the importance of having information on RFR to guide the patient’s management in terms of kidney damage due to the underlying diseases and the use of toxic substances such as the iron chelation therapy. In this particular patient population, the use of biomarkers for the diagnosis of AKI [13,14,15] as used in the ICU population becomes difficult and not applicable in the common clinical practice. Indeed, the RFR assessment could identify patients at higher risk of AKI as the difference between the glomerular filtration rate in the resting state and at maximum capacity could provide additional information on kidney health and renal function prognosis. In fact, kidney function is dynamic and continuously adjusts to changes in the internal environment to maintain homeostasis. The GFR increases in response to various physiological and pathological stressors, including oral protein intake. Despite longstanding interest in RFR as a biomarker in nephrology, its underlying mechanisms remain inadequately understood. Previous studies on RFR have used various measurement methods and yielded heterogeneous results [16]. In the past, the oral protein load was used as it is generally accepted that animal protein ingestion can induce a large increase in GFR by modifications in renal hemodynamics [17,18]. The degree of GFR increase in response to kidney hemodynamics self-regulation represents the RFR that can be calculated as the difference between the maximal creatinine clearance detected during the test and the basal one. Unfortunately, in clinical practice, the protein load is not so easy to perform; rather, it requires an 8-h observation period for the patient’s assessment reducing the application and reproducibility of this test in clinical practice.

Recently, an ultrasound RFR analysis test was validated using an ultrasound and Doppler technique in a population of healthy volunteers [19] and subsequently validated to assess the occurrence of subclinical postoperative AKI in heart surgery [20]. This ultrasound and Doppler test, known as Renal Stress Test (RST), is based on the measurement of the percentage change in the intraparenchymal renal resistive index (IRRIV) between the basal status in rest condition (Figure 1a) and after an external stress (Figure 1b). The stress consists in the application of a pressure at the level of the abdominal wall. This pressure is generated by the application of a weight on the abdomen wall corresponding to 10% of the patient’s body weight (Figure 1b), as reported by Samoni et al. [19,21]. The abdominal compression compresses the kidney vessels, determining a reduction in the renal perfusion and the induction of the kidney hemodynamics self-regulation mechanism with afferent arteriole vasodilatation that can be detected and quantified by a variation in the renal resistive index (RRI) during the stress compared to baseline value [22,23]. Oral protein loading and RST share the same mechanism of kidney hemodynamics self-regulation to control the GFR and therefore the RST can be used for the RFR assessment as previously reported in the literature [19]. According to the literature, the RST can be applied in clinical practice as a routine test to assess the presence of RFR in selected populations and in order to assess the susceptibility to AKI for the best patients’ management. A standardized and clinically feasible approach to quantifying RFR would allow for more rigorous appraisal of its value as a biomarker and could pave the way for adoption of “renal stress tests” into clinical practice.

The aim of this pilot study is to apply the RST in a β-TM population to evaluate the RFR presence and correlate it to the iron overload and to provide a guide parameter in assessing patients’ renal injury and their susceptibility to AKI.

## 2. Materials and Methods

### 2.1. Subjects

From 1 October 2021 to 1 July 2022 the data of all patients undergoing the RST for the internal good clinical practice at the Nephrology Unit of the University Hospital of Padua were collected retrospectively in order to identify the population for this pilot study. All subjects have given informed consent to the use of their data upon acceptance of the ultrasound execution. Patients affected by β-TM were considered eligible for the study. Inclusion criteria were: (a) age more than 12 years old, (b) diagnosis of β-TM, (c) baseline estimated GFR, calculated using the CKD-EPI equation, higher than 90 mL/min/1.73 m^2^. Exclusion criteria were: (a) chronic therapy that may modify renal blood flow and/or GFR (Angiotensin Converting Enzyme-Inhibitors, Angiotensin Receptor Blockers, beta-blockers, calcium channel blockers, loop diuretics etc.) and/or (b) nonsteroidal anti-inflammatory drugs (NSAIDs) in the 2 days before the tests, (c) ultrasound evidence of morphological kidney abnormalities and/or renal artery stenosis, (d) atrial fibrillation, (e) hypertension, (f) history of primitive glomerular disease and (g) ascertained urinary infection, (h) fever in the past 7 days, (i) reduced water intake with possible dehydration.

### 2.2. Baseline RRI Measurement

The RRI measurement was performed by one trained sonographer using a multi-frequency convex probe and the same ultrasound machine in controlled environmental conditions (HI Vision Ascendus—Hitachi with probe C715 model—constant ambient temperature of 21 °C). RST was performed in the patient in supine position after a rest period of at least 15 min. RRIs were manually measured for all patients excluding the automatic software. The RRI were calculated by the built-in software as the following formula: RRI = [(peak systolic velocity − end diastolic velocity)/peak systolic velocity], in which peak systolic velocity and end diastolic velocity were measured in the same wave (Figure 2). The basal RRIs were measured on superior, middle, and inferior interlobular arteries in each kidney and the mean value was considered as the baseline value (baseline RRI). Both kidneys were analyzed to identify the mean baseline value. If there were no differences greater than 0.02 between the two mean values, the test was performed on the right kidney only. In our population, all patients presented a difference ≤0.02 in the basal value of RRI.

### 2.3. Renal Stress Test (RST)

RST was performed in all subjects as a standard ultrasound for the best clinical practice. After a rest of at least 15 min in supine position, the patient was studied for the baseline RRIs. Then, a saline bag of 10% of the patient’s body weight was placed on the anterior abdomen wall to induce the kidney stress reducing the renal perfusion, stimulating the kidney self-regulation mechanism to maintain the GFR by afferent artery vasodilation. The sonographer recorded the RRIs in a middle interlobular artery each minute for 10 min during this stress in order to assess the change in RRIs related to the self-regulation mechanism (Figure 3). The Intra Renal Resistive Index Variation (IRRIV) was defined as the percentage difference between the value of the baseline RRI and the smallest RRI value detected during the test, while the absolute change in IR was express as Delta RRI (baseline RRI–minimum RRI value during stress). We define a positive RST when the Delta RRI is >0.05 as qualitative parameter [19], while the amplitude of IRRIV and its standardization for body surface area (IRRIV_SC_) describe the RST as quantitative parameters of RFR. In details, as previously described by the literature [19], the Delta RRI value was considered significant only if higher than 0.05, according to CD indirect criteria of renal artery stenosis diagnosis utilized by some authors [24].

### 2.4. Statistical Analysis

Data are reported as means and standard deviations (SD). Categorical variables are reported as absolute and relative frequencies. The correlations between data were tested using Pearson correlation analysis.

ANOVA was used to compare quantitative variables between groups and Student’s *t*-test for paired variables.

Values of 5% or less (*p* < 0.05) were considered significant.

## 3. Results

Nineteen β-TM patients were enrolled in this study. The characteristics of subjects at baseline and during the RST are shown in Table 1.

The mean baseline creatinine (sCr) was 50 ± 11 umol/L, corresponding to an estimated baseline GFR (eGFR) of 130 ± 19 mL/min/1.73 m^2^ (CKD-EPI equation). The average ferritin value of the last year was 1282 ± 723 ng/mL. Fourteen patients had tubulopathy during their medical history defined as the presence of proteinuria, acute tubular necrosis or Fanconi syndrome. All patients were under iron chelation therapy: five patients in deferiprone (DFP), five in deferoxamine (DFO) and nine patients in deferasirox (DFX). Table 1 shows the RST values: the mean baseline RRI was 0.67 ± 0.06, Delta RRI was 0.075 ± 0.032, IRRIV was 10.9 ± 4.3% and standardized IRRIV for the patient’s body surface was 6.8 ± 2.5%.

In males, the baseline RRI and Delta RRI were significantly higher than female patients (respectively, *p* = 0.03 and *p* = 0.05), as shown in Figure 4 and Figure 5. The study shown an inverse correlation between mean ferritin values and Delta RRI (R = −0.51, *p* = 0.03) as shown in Figure 6, between ferritin and IRRIV values (R = −0.57, *p* = 0.01) as shown in Figure 7 and between ferritin and IRRIV_SC_ (R = −0.47, *p* = 0.04) as shown in Figure 8.

We did not identify any correlation between the RST results, and the type of iron chelation therapy. There is also no correlation between the RST and a positive history for tubulopathy.

## 4. Discussion

Our results suggested that patients with β-TM have a substantially higher baseline RRI than expected values in relation to low creatinine levels of 50 ± 11 umol/L and high eGFR of 130 ± 19 mL/min/1.73 m^2^. In the literature, RRIs’ normal values are between 0.60–0.70 [25], with 0.70 considered to be the upper normal threshold in adults [21]. Therefore, it seems that in this population the hyperfiltration determines elevated eGFR values and reduced serum creatinine, while the underlying β-TM can induce subclinical (and then clinical) kidney injury with an increase in basal RRI. In this cohort, male patients have statistically higher values than females. Furthermore, in 18/19 subjects, the RST can be considered positive as the Delta RRI > 0.05 indicates the RFR presence [20]. Furthermore, we found that the mean ferritin values, which is the clinical expression of the Iron overload, of β-TM patients are inversely related to RST, percentage of IRRIV and IRRIV_SC_. This data led us to hypothesize that in patients with iron overload, the RFR is present but the lower Delta RRI, IRRIV and IRRIV_SC_ values indicate its reduced amplitude, probably due to the renal damage induced by the β-TM [25].

β-TM patients also present changes in glomerular filtration due to anemia that is thought to reduce systemic vascular resistance, causing an increase in renal plasma flow and GFR [10]. In fact, in our population the GFR values are increased with evidence of glomerular hyperfiltration, as demonstrated by the eGFR. It is known that glomerular hyperfiltration can be harmful to the mesangial compartment, causing, over time, sclerotic processes and renal damage [26,27,28] that can induce an increase in RRI. The glomerular hyperfiltration and kidney injury can induce a different degree of tubular dysfunction, determining proteinuria (8.6%), hypercalciuria (12.9%), phosphaturia (9.2%), hyperuricosuria (38%), magnesiuria (8.6%), and increased excretion of β2-microglobin (β2Μ) (13.5%) [29]. Tubular dysfunction is most likely due to iron overload, with hyperactivation of the oxidative stress cascade and reduction in NO production, chronic anemia and the use of iron chelators [9,30,31,32]. Non-transferrin iron can lead to organelle membrane dysfunction and subsequent cell injury and death. Iron-catalysis generation of reactive oxygen species (ROS) is responsible for initiating the peroxidative reaction [33] and the chronic iron deposition in proximal tubules, glomeruli and renal interstitium was associated with significant glomerulosclerosis, tubular atrophy, and interstitial fibrosis [32]. All of these mechanisms can contribute to the injury of the renal parenchyma with progressive reduction of the RFR as detected by progressive lower Delta RRI, IRRIV and IRRIV_SC_ values. This RFR reduction can first be subclinical with no alterations in the basal creatinine levels and eGFR, and then become clinically evident with an increase in creatinine levels, determining the loss of more than 50% of the renal parenchyma. Interestingly, β-TM patients have a condition of hyperfiltration linked to anemia and, at the same time, a tubular damage due to the iron overload and iron chelators. These two factors can determine a balance in which hyperfiltration compensates for kidney damage until there is a loss of at least 50% of nephron mass. In this situation, even with normal creatinine levels, there is a progressive reduction in RFR that currently cannot be identified using only the plasma creatinine level and the eGFR as occurs in common clinical practice.

According to our preliminary data in β-TM patients, the positive RST indicates the presence of RFR, but the progressive mesangial and tubular injury can be responsible for hemodynamic and functional kidney rearrangement, thus determining the higher baseline RRI. Finally, the RFR could be affected by iron overload as demonstrated by the reverse relation between ferritin higher levels and the lower Delta RRI, IRRIV and IRRIVsc.

A total of 1/19 patients did not present RFR in an history of severe tubulopathy. In this situation, the RST could not detect RFR, and the presence of anemia-induced hyperfiltration kept creatinine and eGFR levels within the normal range despite the presence of kidney damage, masking this kidney injury under an apparent renal normal function.

The results of this pilot study suggested that the RST can be used to assess the RFR during the β-TM patient’s follow-up to identify clinical conditions that can reduce the RFR such as the β-TM renal complications and the toxic effects of the iron chelation therapy.

In fact, it is known from the literature that RFR measurement may reveal subclinical loss of renal function, early phases of CKD and a patient’s high susceptibility to toxic exposures with AKI [18].

Our pilot study showed an inverse correlation between ferritin values and RST in a population affected by β-TM by a reproducible, inexpensive and safe ultrasound and Doppler test. In conclusion, in β-TM the RST can provide useful clinical information to manage and modify the therapeutic approach to detect, early on, subclinical or clinical reduction in the RFR with higher predisposition to AKI with evolution in CKD over time.

A limitation of this study is the limited number of patients and the monocentric population. Further studies in larger and multicentric populations are necessary to better understand the role of the different iron chelators and the tubulopathy history in the kidney damage.

## 5. Conclusions

The RST is a safe, reproducible, and inexpensive ultrasound tool to identify the RFR in those individuals who have normal creatinine and eGFR values that can present a kidney injury due to an underlying disease such as the β-TM. The RST seem to allow for an early identification of those patients who may be predisposed to a subclinical or a clinical AKI in order to modify the medical therapy and provide clinical management tailored to the patient.

## Figures and Tables

**Figure 1 jcm-11-06752-f001:**
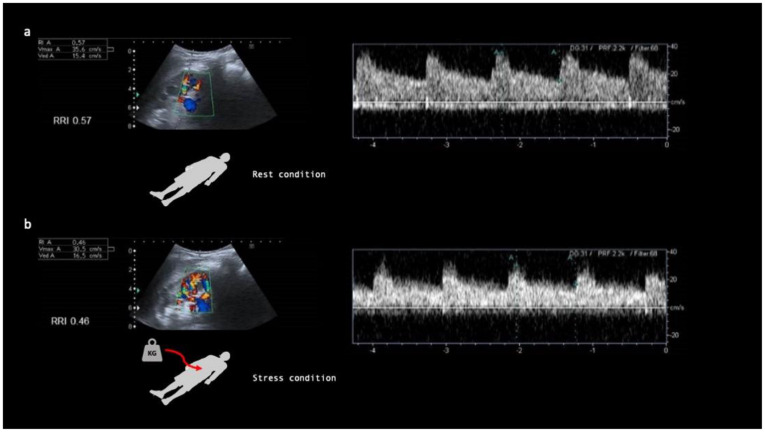
Renal Stress Test (RST) is based on the measurement of the percentage change in the intraparenchymal renal resistive index (IRRIV) between the basal status in rest condition (**a**) and after an external stress to study the RFR (**b**). In resting conditions (**a**), the RRIs are measured by studying the average value which is then used to calculate the percentage and absolute variation with respect to the values found after the stress (**b**) applied to the kidney through the application of the weight on the anterior abdominal wall. The figure shows the sampling points of the Doppler signal in the kidney and the relative Doppler spectra that are used for the calculation of the RRI.

**Figure 2 jcm-11-06752-f002:**
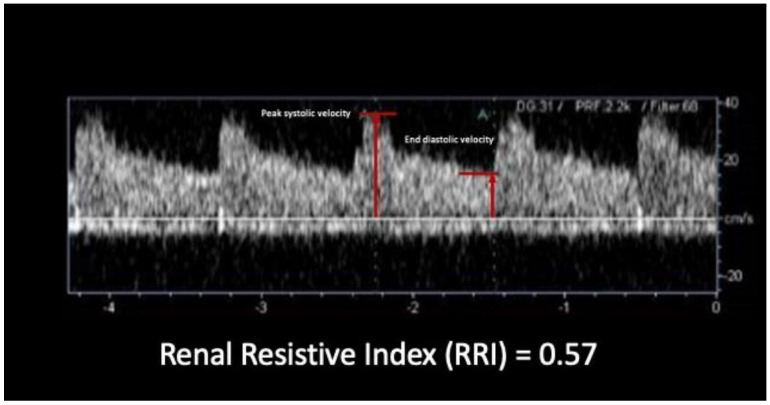
RRI calculation in the Doppler analysis. The figure shows a typical low resistance Doppler spectrum in a renal interlobar artery. The markers highlighted are related to the maximum systolic peak and to the end-diastolic velocity which are used for the calculation of the RRI. Both markers are measured in the same cardiac cycle.

**Figure 3 jcm-11-06752-f003:**
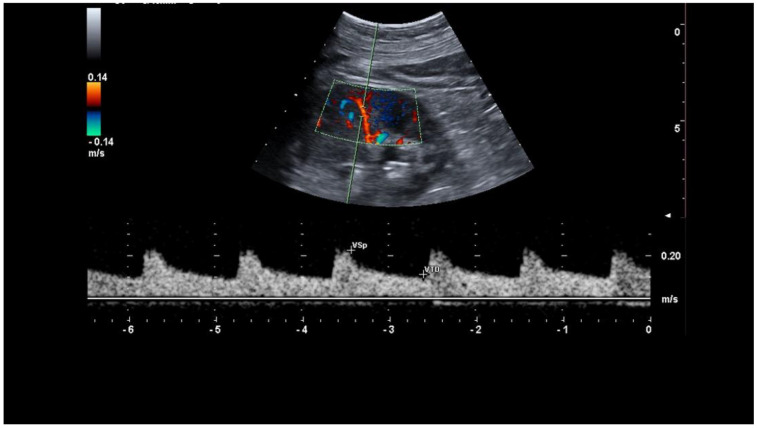
RRIs in a middle interlobular artery. The figure shows the exact sampling point of the Doppler signal at the level of the interlobar artery of the kidney for the representation of the Doppler spectrum on which the RRI is calculated. To note the regularity of the cardiac cycle (sinus rhythm) which allows to have an accurate measurement of the RRI and the sampling of the peak systolic and end diastole velocity in the same cardiac cycle.

**Figure 4 jcm-11-06752-f004:**
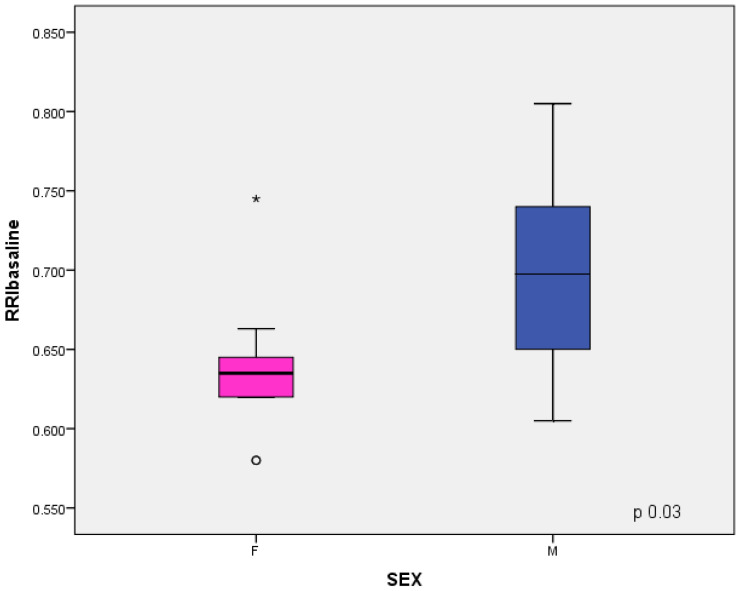
RRI in male and female patients. Box plot showing the differences of RRI baseline between male and female subsamples. Comparison with Student’s *t*-test for independent samples showed that males had higher baseline RRI values than females. * and ○ are two extreme values.

**Figure 5 jcm-11-06752-f005:**
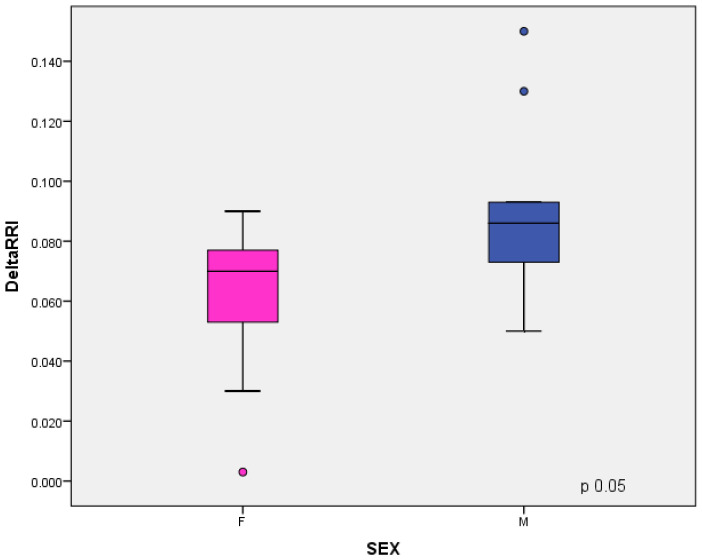
Delta RRI in male and female patients. Box plot showing the differences of Delta RRI between male and female subsamples. Comparison with Student’s *t*-test for independent samples showed that males had higher Delta RRI values than females.

**Figure 6 jcm-11-06752-f006:**
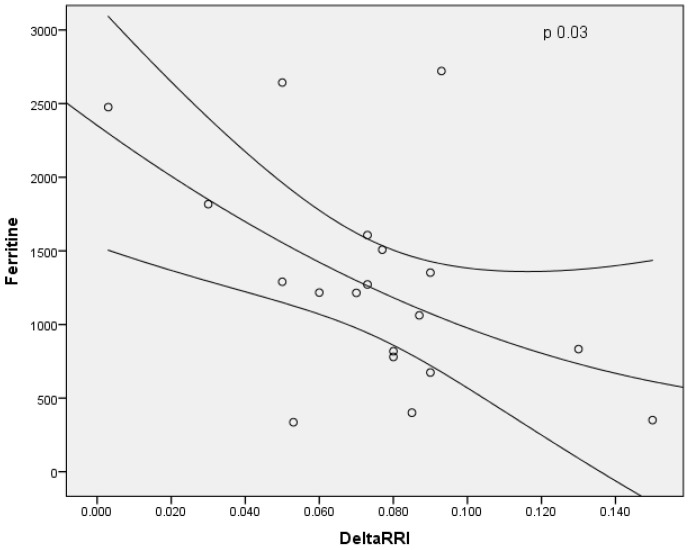
Inverse correlation between mean ferritin values and Delta RRI. Dot plot illustrating the correlation between mean ferritin values and Delta RRI (R = −0.51, *p* = 0.03).

**Figure 7 jcm-11-06752-f007:**
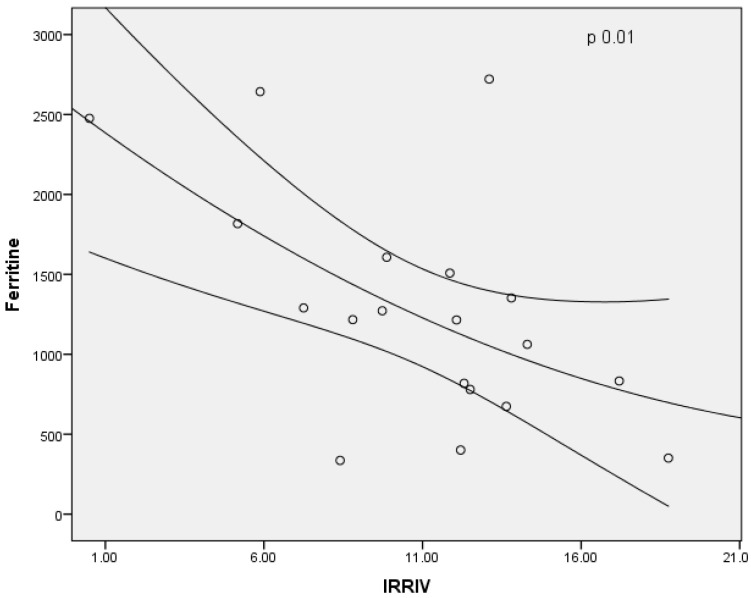
Inverse correlation between ferritin and IRRIV values. Dot plot illustrating the correlation between men ferritin and IRRIV values (R = −0.57, *p* = 0.01).

**Figure 8 jcm-11-06752-f008:**
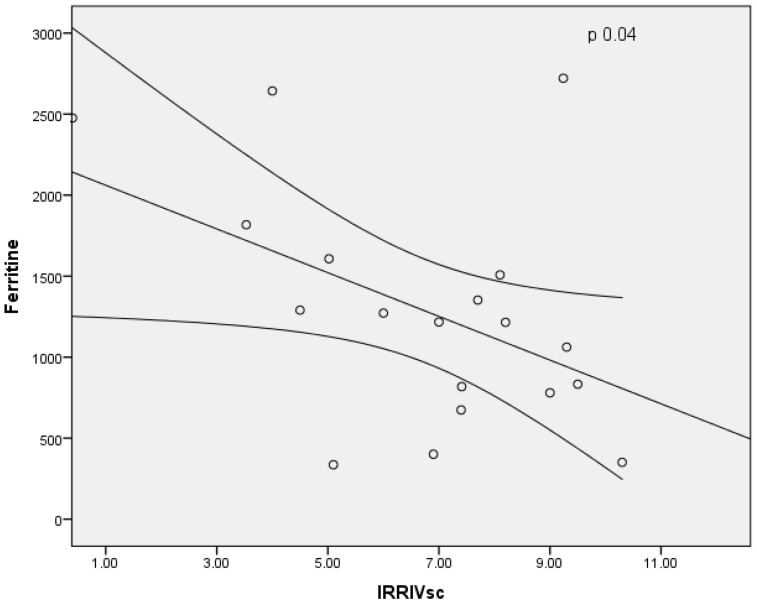
Inverse correlation between ferritin and IRRIV_SC_. Dot plot illustrating the correlation ferritin and IRRIV values standardized for body surface area (R = −0.47, *p* = 0.04).

**Table 1 jcm-11-06752-t001:** Characteristics of patients at baseline and during RST.

Baseline Data	Entire Cohort (*n* = 19)
Sex (male)	10
Age (years)	33 ± 16
Height (cm)	159 ± 9
Weight (Kg)	57.9 ± 12.6
BMI (Kg/m^2^)	22.8 ± 3.6
BSA	1.73 ± 0.43
Baseline sCr (umol/L)	50 ± 11
Baseline eGFR (mL/min/1.73 m^2^)	130 ± 19
Average Ferritin in last year (ng/mL)	1282 ± 723
Deferiprone (number of patient)	5
Deferoxamine (number of patient)	5
Deferasirox (number of patient)	9
Tubulophaty * (number of patient) * proteinuria, acute tubular necrosis or Fanconi syndrome	14/19
**RRI baseline**	0.67 ± 0.06
**Delta RRI**	0.075 ± 0.033
**IRRIV (%)**	10.9 ± 4.3
**IRRIV SC (%)**	6.8 ± 2.5

## Data Availability

Not applicable.

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
