# Peer review of "The Assessment of Renal Functional Reserve in β-Thalassemia Major Patients by an Innovative Ultrasound and Doppler Technique: A Pilot Study"

_jcm, 2022, doi:10.3390/jcm11226752_

Round 1

Reviewer 1 Report

In the current study titled “The assessment of Renal Functional Reserve in β-thalassemia

major patients by an innovative ultrasound and Doppler technique: A pilot study” by Nalesso et.al. authors have addressed application of ultrasound and doppler to quantify RFR by measuring RST in b-Thalassemia patients. The method itself has been used earlier for measurements in healthy individuals (PMID: 26649340- Not addressed in the current manuscript) but no further articles published in this area and, current novelty is to being utilized the technique for b-Thalassemia patients.

The manuscript is well organized in terms of relevant literature, background, result and discussion. However, I strongly recommend revising the introduction and making it more concise as, with such an elaborated introduction a connect of transitioning throughout is missing and making such an interesting article less reader friendly moreover, it tough to address main highlight of the overall hypothesis. The methods are apt, and results supports the hypothesis and mechanism also has been illustrated well.

Introduction-

Minor:

Line 101: Minor spell check required.

Major:

The overall article is innovative and novel finding but, my suggestion to concise to original subject line is missing. Certain literatures to address the techniques used earlier for Kidney function analysis, Kidney function and correlation to b-Thalassemia and how the current method could enhance the current diagnostic modalities should be story line to highlight. So, please address literature emphasizing these would be great (there are quite a few good article available).

Discussion:

No major comments but, I am interested to know if there are any future studies planned for futher evaluation as, n=19 is good for pilot study but need to be tested further on a larger population.

References: 

All relevant and in sync. A couple of them can be added further as addressed in introduction section.

Figure Legends:

Minor:

Adding statistical method/and more text elaboration in the figure legend would be appreciated.

Author Response

REVIEWER # 1

In the current study titled “The assessment of Renal Functional Reserve in β-thalassemia

major patients by an innovative ultrasound and Doppler technique: A pilot study” by Nalesso et.al. authors have addressed application of ultrasound and doppler to quantify RFR by measuring RST in b-Thalassemia patients. The method itself has been used earlier for measurements in healthy individuals (PMID: 26649340- Not addressed in the current manuscript) but no further articles published in this area and, current novelty is to being utilized the technique for b-Thalassemia patients.

The manuscript is well organized in terms of relevant literature, background, result and discussion. However, I strongly recommend revising the introduction and making it more concise as, with such an elaborated introduction a connect of transitioning throughout is missing and making such an interesting article less reader friendly moreover, it tough to address main highlight of the overall hypothesis. The methods are apt, and results supports the hypothesis and mechanism also has been illustrated well.

The introduction was rewritten according to the useful and interesting indications of the auditor.

Introduction

Minor:

Line 101: Minor spell check required. Errors have been identified and corrected.

Major:

The overall article is innovative and novel finding but, my suggestion to concise to original subject line is missing. Certain literatures to address the techniques used earlier for Kidney function analysis, Kidney function and correlation to b-Thalassemia and how the current method could enhance the current diagnostic modalities should be story line to highlight. So, please address literature emphasizing these would be great (there are quite a few good article available).

To introduce the concept and not make the introduction longer, as previously reported by the reviewer, we have included in the text some lines relating to a review on "Renal Functional Reserve Revisited." In this way, readers can deepen the topic and draw food for thought. after reading the whole article.

Palsson R, Waikar SS. Renal Functional Reserve Revisited. Adv Chronic Kidney Dis. 2018 May;25(3):e1-e8. doi: 10.1053/j.ackd.2018.03.001. PMID: 29793670.

Discussion:

No major comments but, I am interested to know if there are any future studies planned for futher evaluation as, n=19 is good for pilot study but need to be tested further on a larger population.

The study presented in this article is a pilot study, we are routinely continuing to perform the test on all patients with thalassemia in our hospital.

References: 

All relevant and in sync. A couple of them can be added further as addressed in introduction section.

We reviewed the literature and added what we found useful for the article.

Figure Legends:

Minor:

Adding statistical method/and more text elaboration in the figure legend would be appreciated. We have implemented the point as requested by the reviewer.

Reviewer 2 Report

Federico Nalesso et al. performed a pilot study to assess renal functional reserve in β-thalassemia major patients by an innovative ultrasound and Doppler technique. However, there are some major concerns.

Major comments

1.        Sample group: In this study, authors enrolled 19 β-TM patients undergoing text and showed in table 1. Why no matched control group data?  How they get the conclusion of β-TM have a substantially higher baseline RRI than expected values in relation to low creatinine and high eGFR?

2.        The iron overload and chronic anemia is related to CKD, so is there relation between RRI and anemia?

3.        The sample size of study is not enough to get the conclusion. And meanwhile, the conclusion lack of clinical significance.

Minor comments

1.The figure is not clear, and without label of p value.

2.the size and typeface of reference need to be modified.

Author Response

Federico Nalesso et al. performed a pilot study to assess renal functional reserve in β-thalassemia major patients by an innovative ultrasound and Doppler technique. However, there are some major concerns. 

Major comments

  1. Sample group: In this study, authors enrolled 19 β-TM patients undergoing text and showed in table 1.
    1. Why no matched control group data?  

Our report is essentially a pilot study. This study is the first step in exploring [doi: 10.1016/j.jpsychires.2010.10.008]Renal Functional Reserve (RFR) evaluated by ultrasound and Doppler technique in β-TM patients. We hypothesized iron overload and related tubulopathy negatively influence  RFR even in patients without evidence of CKD.  As the first step in this field, we chose to study only β-TM patients to establish examination procedures, the feasibility of the method, and the possible influence of iron overload. Finally, we considered the difficulties of having an adequate control group:  healthy people do not have hyperfiltration, anaemic CKD patients have  RFR failure by definition, and β-TM patients without iron overload are uncommon. In this scenario, we prefered to evaluate deeply technical aspects, and results of RST to better intercept an adequate control group.

  1. How they get the conclusion of β-TM have a substantially higher baseline RRI than expected values in relation to low creatinine and high eGFR? 

The RRI values ​​correlate with GFR and therefore serum creatine: in detail in renal norm-function the RRIs are in the normal range with creatinine and GFR values ​​within the limits (the upper is 0.70). In the course of CKD the GFR is reduced due to renal damage and creatinine increases, this condition is associated with higher RRIs above the normal limit  as reported in the literature (for gender and age). In CKD, the destruction of the parenchyma results in an increase in RRIs. In the study, patients have RRIs that are higher than those typically reported in the literature with lower creatinine resulting from an increase in GFR due to anemia. Renal damage is partially compensated by the vasodilation induced by anemia at the level of the afferent renal arterioles which determine an increase in plasma flow with hyperfiltration and increase in GFR which reduces serum creatinine even in the presence of damage. This mechanism of hyperfiltration over time leads to a progressive destruction of the renal parenchyma and is the basis of the pathophysiology of the progression of CKD in all diseases that are capable of causing renal damage over time. A control group was not considered as the article is a pilot study and wanted to investigate the preliminary results of this test on a small population of patients with thalassemia. In the (ongoing) larger population study the control group will be considered.

  1. The iron overload and chronic anemia is related to CKD, so is there relation between RRI and anemia?

Iron overload and chronic anemia are related to CKD. Anemia induces a state of systemic vasodilation which can also affect the renal vascularization with repercussions on the GFR which increases and therefore on the RRIs which are reduced as described in the text. These data are supported by the literature as reported in the article.

  1. The sample size of study is not enough to get the conclusion. And meanwhile, the conclusion lack of clinical significance.

 We are agreeing with the revision, the aim of a pilot study is not to get any conclusion, especially a clinically significant conclusion. Our preliminary results show an interesting correlation between delta RRI and ferritin, but obviously, any possible interpretation should be tested and validated by future studies in this field.

To stress this aspect, we deeply revised the text, emphasizing the fact we do not have a conclusion, but we have a hypothesis to test. 

Minor comments

1.The figure is not clear, and without label of p value. The figures have been corrected according to the indications of the reviewer.

2.the size and typeface of reference need to be modified. The required corrections have been made.

Reviewer 3 Report

I find this paper very interesting and innovative. I have no major complaints except for figures 4,5,6,7 and 8. The quality of the images is unsatisfactory. Also, new and more modern charts should improve overall quality of this scientific paper. Figures must be described in more details.

Author Response

I find this paper very interesting and innovative. I have no major complaints except for figures 4,5,6,7 and 8. The quality of the images is unsatisfactory. Also, new and more modern charts should improve overall quality of this scientific paper. Figures must be described in more details.

We have proceeded with the corrections in the article as per the indications given by the reviewer.

Reviewer 4 Report

I have reviewed Nalesso F., et al. manuscript entitled "The assessment of Renal Functional Reserve in β-thalassemia 2 major patients by an innovative ultrasound and Doppler tech-3 nique: A pilot study." They demonstrated that Ultrasound assessment of intra-parenchymal renal resistive index variation (IRRIV) can be used to identify the RFR presence when the Delta Renal Resistive Index (RRI) is > 0,05  in the Renal Stress Test (RST). Besides, they also found an inverse correlation between ferritin values and the RFR. their findings demonstrated the RST ability to assess the RFR by ultrasound, which was with a clinical significance to guide the β-TM patient’s management and iron chelation therapy.

Overall, their research has some clinical implications, and there are a few minor issues in the manuscript.

1. please define the CKD abbreviation in first mention (line 62 but not 69). 

2. One more space in front of sentence "The ap-plication of RST in this population allows to integrate the test in the clinical practice to provide a guide parameter in assessing patients' renal injury and their susceptibility to AKI." in the line 112.

3. In conclusion, is it a typing mistake about "RTS"? I think it should be "RST". (line 279)

Author Response

I have reviewed Nalesso F., et al. manuscript entitled "The assessment of Renal Functional Reserve in β-thalassemia 2 major patients by an innovative ultrasound and Doppler tech-3 nique: A pilot study." They demonstrated that Ultrasound assessment of intra-parenchymal renal resistive index variation (IRRIV) can be used to identify the RFR presence when the Delta Renal Resistive Index (RRI) is > 0,05  in the Renal Stress Test (RST). Besides, they also found an inverse correlation between ferritin values and the RFR. their findings demonstrated the RST ability to assess the RFR by ultrasound, which was with a clinical significance to guide the β-TM patient’s management and iron chelation therapy.

Overall, their research has some clinical implications, and there are a few minor issues in the manuscript.

  1. please define the CKD abbreviation in first mention (line 62 but not 69). 

The correction was made as indicated by the reviewer.

  1. One more space in front of sentence "The ap-plication of RST in this population allows to integrate the test in the clinical practice to provide a guide parameter in assessing patients' renal injury and their susceptibility to AKI." in the line 112.

The correction was made as indicated by the reviewer.

  1. In conclusion, is it a typing mistake about "RTS"? I think it should be "RST". (line 279)

The correction was made as indicated by the reviewer.

Round 2

Reviewer 2 Report

It is ok for publishment